# Sleep and COVID-19. A Case Report of a Mild COVID-19 Patient Monitored by Consumer-Targeted Sleep Wearables

**DOI:** 10.3390/s21237944

**Published:** 2021-11-28

**Authors:** Arnaud Metlaine, Fabien Sauvet, Mounir Chennaoui, Damien Leger, Maxime Elbaz

**Affiliations:** 1EA 7330 VIFASOM, Université de Paris, 1 Place du Parvis notre Dame, 75004 Paris, France; arnaud.metlaine@aphp.fr (A.M.); fabien.sauvet@gmail.com (F.S.); mounirchennaoui@gmail.com (M.C.); damien.leger@aphp.fr (D.L.); 2Centre du Sommeil et de la Vigilance, Hôtel-Dieu, AP-HP, 1 Place du Parvis notre Dame, 75004 Paris, France; 3Institut de Recherche Biomédicale des Armées (IRBA), 1 Place du Général Valérie André, 91190 Paris, France

**Keywords:** COVID-19, consumer sleep wearables, sleep

## Abstract

Since its first description in Wuhan, China, the novel Coronavirus (SARS-CoV-2) has spread rapidly around the world. The management of this major pandemic requires a close coordination between clinicians, scientists, and public health services in order to detect and promptly treat patients needing intensive care. The development of consumer wearable monitoring devices offers physicians new opportunities for the continuous monitoring of patients at home. This clinical case presents an original description of 55 days of SARS-CoV-2-induced physiological changes in a patient who routinely uses sleep-monitoring devices. We observed that sleep was specifically affected during COVID-19 (Total Sleep time, TST, and Wake after sleep onset, WASO), within a seemingly bidirectional manner. Sleep status prior to infection (e.g., chronic sleep deprivation or sleep disorders) may affect disease progression, and sleep could be considered as a biomarker of interest for monitoring COVID-19 progression. The use of habitual data represents an opportunity to evaluate pathologic states and improve clinical care.

## 1. Introduction

COVID-19 is a potentially severe respiratory infection caused by the SARS-CoV-2 virus, first identified in Wuhan, China in December 2019. The DNA sequence was rapidly made public and numerous research studies have followed. SARS-CoV-2 is mainly transmitted by droplets and aerosols from asymptomatic and symptomatic infected subjects. The consensus estimates for the basis reproduction number (R0) ranges between 2 and 3, and the median incubation period is 5.7 (range 2–14) days. The pandemic remains active to this day with a worldwide death toll of over 5,012,337 [1]. While most cases are mild, 5–10% of patients are hospitalized, mainly due to pneumonia with severe inflammation or acute respiratory distress syndrome. Complications are respiratory and multiorgan failure; risk factors for complicated disease are higher age, hypertension, diabetes, chronic cardiovascular, chronic pulmonary disease, and immunodeficiency. The current estimate for the infection’s fatality rate is 0.5–1%, and the prediction of severe forms of the disease is still a challenge for the physician [2].

The SARS-CoV-2 pandemic has been marked by the development of the use of ambulatory medical devices and nonmedical wearables for the monitoring of patients in ambulatory settings. While the diffusion of these technologies has great potential for the production of health-related information, it is important to evaluate the way the data can be used in the medical decision-making process [3].

In this paper the evolution of SARS-CoV-2-related sleep disorders and physiological parameters are examined in a SARS-CoV-2 infected patient who was routinely wearing three consumer sleep wearables before the onset of the disease and kept them throughout disease and recovery. This clinical case illustrates the clinical interest of routine wearable-generated data in assessing health status during an acute and potentially serious infectious disease.

## 2. Materials and Methods 

### 2.1. Clinical Case

On 18 March 2020, a 53-year-old male patient with a body mass index (BMI) of 25 kg/m^2^ presented with a mild case of COVID-19. Apart from a history of asthma, the patient was a healthy nonsmoker subject. On day 1, symptoms started with a 38 °C fever, intense headache, digestive discomfort, and intestinal urgency with cough. These symptoms appeared 5 days after prolonged contact (one-hour meeting with less than 1 m distancing and no protective measures) with a work colleague who afterwards tested positive for COVID-19 as well. On day 2, the situation deteriorated quickly with a cough, worsening headache, and diarrhea. Fatigue, dizziness, and joint pain (back and lower limbs) developed on day 3. On day 4, the fever ranged between 38.2 °C and 38.5 °C. The patient was self-medicated with paracetamol. RT-PCR test returned positive on day 5. The patient remained in bed most of the day with severe lethargy and fits of dry cough (days 6 to 8). One day 9, a thoracic CT-scan revealed small pulmonary lesions consistent with COVID-19. The patient was prescribed inhaled budesonid (200 µg) and formoterol (6 µg) per day and oral Azithromycin 500 mg (1 day) and 250 mg (4 days). On day 10, he experienced breathing difficulties with chest pressure with a SpO2 level at home of 93%. The patient was transferred to the emergency services where blood test results were found to be within normal limits, except for leucopenia [WBC: 3.310 (per µL)], lymphopenia [1.020 (per µL)], and elevated CRP [12.3 mg/L]. On day 11, the patient experienced complete anosmia and ageusia for 5 days. The disease evolution was characterized by fatigue requiring prolonged rest, and spikes of fever up to 39.8 °C. On day 16, the subject recovered the sense of taste and smell, and he recovered completely from all symptoms by day 23. The subject lost 7 kg over 13 days. On day 25, serology showed positive IgG and IgM results indicative of past infection, and a second RT-PCR test on day 28 returned negative. 

### 2.2. Subjective Sleep Data: From Incubation Phase to Recovery

Before infection, the patient was considered as a good sleeper with an acute sleep deprivation (TST less than 6 h over the ten days prior to infection). Upon COVID-19 onset, the quality of sleep decreased rapidly. The patient, confined to his home, lost his usual sleep pattern. From day 1 to 9, the patient described continuous worsening of night sleep (initiation and maintenance sleep disorders), with an unusual afternoon nap. Progressively, sleep duration increased with times ranging from 10 to 12 h per day. Sleep fragmentation increased into days 10–18. The recovery of sleep was slow characterized by a progressive reduction in sleep duration, which resumed to 7 h on day 19. Sleep quality improved 2 days before normalization of taste and smell. One month later the subject reported good quality and efficiency of sleep and a return to his usual pattern (TST around 6.30 to 7 h).

### 2.3. Objective Sleep Data Assessment 

The patient voluntarily recorded his sleep and wake rhythms via three consumer sleep wearables (CSW): Oura ring Gen 2 (Oura) [4], Fitbit Versa 2 (Fitbit now part of Google) [5], and iSleep Watch for AppleWatch (iSommeil) [6]. FitBit Versa 2 watch and iSleep Watch were worn alternatively on the nondominant wrist for 55 days. Oura ring was worn in real life continuously on the nondominant finger for 55 days. Those three CSW (Table 1) had heart rate sensors and 3 axis-accelerometers to estimate sleep duration (TST), WASO. Additionally, the Fitbit Versa 2 watch recorded respiratory rate, and the Oura ring recorded skin temperature and respiratory rate.

Physiological data were recorded as per the following protocol: (1) the Oura Ring was worn continuously; (2) the Apple Watch and the FitBit Versa 2 were worn in an alternating manner, 12 h each. 

All CSW were Bluetooth-compatible and synchronized to an iPhone XS Max patient (Apple, Inc.; Cupertino, CA, USA). Oura ring (Oura) and Fitbit Versa 2 (Fitbit now part of Google) provided the following parameters: sleep stages, total sleep time (TST), time in bed (TIB), wake after sleep onset (WASO), sleep efficiency (SE), sleep latency (SL), and resting heart rate (RHR). iSleep Watch by iSommeil provided TST, TIB, WASO, SE, and RHR. Body skin temperature was calculated automatically by Oura algorithm. We aggregated all the data of the 3 CSW, which were very similar for sleep parameters and for Resting Heart Rate and Breathing Rate. We averaged the data from different devices. All the data were complete.

### 2.4. Data Analysis 

Sleep raw data were available from all three CSW platform.

Statistical tests were performed using R studio (V.0.99.175 2009–2014 RStudio) and significance (α risk) was fixed at *p* < 0.05. Continuous variables are presented as mean ± SD (m ± SD) and means were compared using a repeated measure ANOVA and a post hoc Bonferroni test. Quantitative variables are presented as occurrence and percentage (n (%)). 

Correlations were performed using the Pearson test between quantitative values.

## 3. Results

The mean breathing rate, heart rate, and body temperature increased significantly during the infection (Figure 1) as previously described [7].

We noted that heart rate remained elevated after the active disease period compared to that recorded pre-infection. In addition, the number of steps recorded dropped drastically during COVID-19 (3000 ± 2500 steps versus 9000 ± 2500, *p* < 0.001) (Figure 2A).

Additionally, sleep patterns appeared to be significantly affected (Figure 1). The mean total sleep time increased significantly during the infection (376 ± 86 min versus 472 ± 66 min, *p* < 0.001). Sleep fragmentation, an index of sleep quality, also significantly increased during the infection. In particular the wake after sleep onset cumulative duration (WASO) increased (52 ± 12 versus 80 ± 12 min, *p* < 0.001). We found a significant correlation (*p* < 0.001) between COVID symptoms and sleep parameters. Changes in sleep parameters have been observed as predictors of SARS-CoV-2 infection, independently of physiological changes (Figure 2B).

## 4. Ethics Declarations

The patient provided informed consent prior to engaging in any study procedures. This study was conducted according to the guidelines of Declaration of Helsinki, and approved by the local ethics committee of Ile de France, from Paris, France.

## 5. Discussion

This case study raises interesting points from multiple perspectives. 

First, the study gives a detailed description of the sleep of a mild COVID-19 patient, recorded digitally over 55 days, encompassing the period before infection, incubation, active, and recovery phases. To our knowledge, this is the first sleep study of a patient quarantined at home. It demonstrates that innovative digital tools can have a clinical interest in evaluating patients in real-life conditions. Additionally, it has been shown that data generated by those devices were consistent with those of polysomnography, actigraphy, and self-sleep assessment [4,5,6]. 

Second, little data exist so far on COVID-19 associated sleep patterns. Similar to previous evidence [7], our case study showed a significant reduction in sleep duration (*p* < 0.0001) and an increase in WASO (*p* < 0.0001) in the acute illness phase. Changes in sleep quality appear to be common in minor forms of the infection, and early nonpharmaceutical interventions aiming at improving the sleep of COVID-19 patients are useful [8,9]. 

Third, on day 10, our patient showed significant lymphopenia, which coincided with worsening COVID-19 symptoms and increased sleep fragmentation. COVID-19 patients with lymphopenia are associated with poor sleep and have higher severity than good sleeping COVID-19 patients [8]. Sleep has a modulating role in multiple physiological functions, including immune response [10]. 

We show that wearables could easily be used for remote patient monitoring in mild cases. The high positive correlations between TST, WASO, heart rate, temperature, and COVID-19 symptoms are noteworthy. An early detection algorithm (EDA) that includes sleep parameters should be developed. Wearable devices could be useful in limiting the transmission of infection during a critical period in the disease process by alerting users of a possible early infection [10,11]. Additionally, this case study shows how data from devices they routinely wear could be used for medical purposes constituting a potential paradigm shift in clinical practice.

## 6. Conclusions

This case report highlights the clinical importance of sleep evaluation in COVID-19 patients and how early intervention and management of sleep disorders in the broader population may be recommended to prevent potential sleep-induced frailty in the event of an acute infectious event.

In addition, we suggest wearables could be used for early detection of infection and remote monitoring, allowing patients to report their vital signs from home.

## Figures and Tables

**Figure 1 sensors-21-07944-f001:**
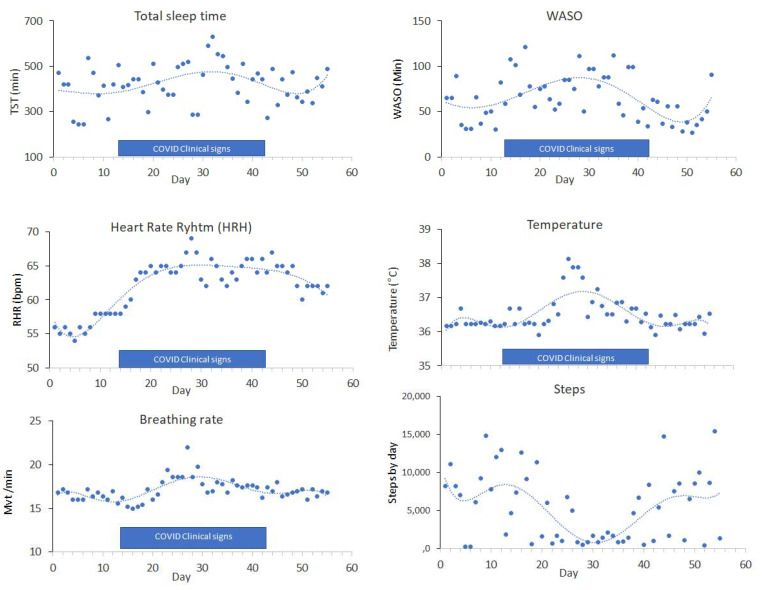
Nocturnal total sleep time, skin temperature, heart rate, breathing, and steps before, during, and after SARS-CoV-2 active disease.

**Figure 2 sensors-21-07944-f002:**
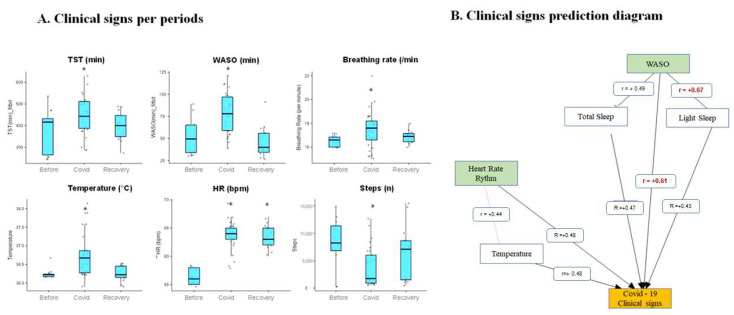
(**A**) Average physiological parameters before, during, and after SARS-CoV-2 (COVID) infection and (**B**) Correlation analysis between TST, WASO, temperature, heart rate rhythm, light sleep, and COVID-19 clinical signs. * is a significant difference with before SARS-CoV-2, (r) are Pearson correlation coefficients between values.

**Table 1 sensors-21-07944-t001:** Device descriptions.

Name	Fitbit Versa 2 (Fitbit Now Part of Google)	Oura Ring (Oura)	iSleepWatch for AppleWatch5 (iSommeil)
Picture	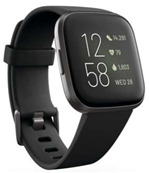	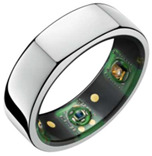	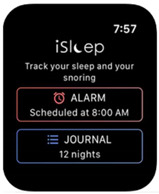
Resting Heart Rate	YES	YES	YES
Body Temperature	NO	YES	NO
Actigraphy 3 axis accelerometer	YES	YES	YES
Respiratory rate	YES	YES	NO
Snoring	NO	NO	YES
Sleep analysis validation reference	Haghayegh et al., 2019 [5]	Altini et al., 2021 [4]	Elbaz et al., 2020 [6]

## Data Availability

The dataset supporting the conclusions of this article is not available due to privacy and ethical reasons.

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
