# Peer review of "Sleep and COVID-19. A Case Report of a Mild COVID-19 Patient Monitored by Consumer-Targeted Sleep Wearables"

_sensors, 2021, doi:10.3390/s21237944_

Round 1

Reviewer 1 Report

Thanks for your revisions concerning my suggestions. I am sorry to know that the patient is one of the co-authors. I must congratulate his complete recovery and your great contribution to providing such a rare but comprehensive experience. Congratulation!

All the best,

Author Response

Dear reviewer,

We thanks a lot for your constructive comments.

Best regards,

Co-Authors

Reviewer 2 Report

This is an interesting case report of a COVID-19 patient continuosly monitoring his sleep by commercially available devices. The results are interesting, but there are a few questions to be answered:

1) Did you obtain a formal consent by the patient to publish anonymously his data? Did you obtain an Ethical Committee approval? These questions can appear redundant, as clearly the patients gave you his data, but the Journal should publish clinical data after approval by an IRB.

2) The different devices used by the patient create some methodological issues. Did you use all the data and how? For 2 devices the patient alternated them each other night, did you average the data from dofferent devices? Were data complete or some data were missing? Why did the patient wear so many devices?

3) There are numerous typos in the text, please correct.

4) it would be appropriate to mention whether similar data have ever been collected in other infectious disease. One can expect temperature and heart rate to be higher, and steps to be much lower during a period of infection, as well as a poorer sleep quality.

Author Response

1) Did you obtain a formal consent by the patient to publish anonymously his data? Did you obtain an Ethical Committee approval? These questions can appear redundant, as clearly the patients gave you his data, but the Journal should publish clinical data after approval by an IRB.

Dear Reviewer,

Thanks a lot for your constructive  review report:

Response 1-

The patient is one of the co-authors and conducted this research by his own initiative. He shared his data to the co-authors wrote a formal consent to publish anonymously his data. We obtained the Ethical Committee approval for this reseach. 

The study was conducted according to the guidelines of the Declaration of Helsinki, and approved by the local Ethical Committee (Comité de Protection des personnes (CPP) Ouest IV, from Nantes, France. We add Ethics consideration from line 179 to line 183.

2) The different devices used by the patient create some methodological issues. Did you use all the data and how? For 2 devices the patient alternated them each other night, did you average the data from different devices? Were data complete or some data were missing? Why did the patient wear so many devices?

Response 2

Physiological data were recorded as per the following protocol : 1) the Oura Ring was worn continuously; 2) the Apple Watch and the FitBit were worn in an alternating manner, 12 hours each. We add this sentence frome line 113 to line 115.

The patient used those 3 devices because he is one of our eHealth Team and he is recording usually the 3 CSW. The patient expert of eHealth (co-author of this research) had the idea to record sleep, skin temperature, heart rate, breathing rate and Steps and wanted to prove that CSW are necessary to follow up COVID-19 in real life. We aggregated all the data of the 3 CSW witch are very similar for  sleep parameters (TST, WASO) and for Heart Rate and Breathing Rate we calculated it with Fitbit algorithm. The patient recorded skin temperature with Oura ring.  We averaged  the data from different devices. All the data were complete. We add those sentences from line 137 to line 139.

3) There are numerous typos in the text, please correct.

Response 3

The authors correct the text and the typos.

4) it would be appropriate to mention whether similar data have ever been collected in other infectious disease. One can expect temperature and heart rate to be higher, and steps to be much lower during a period of infection, as well as a poorer sleep quality.

Response 4:

Similar data have been collected in influenza-like illness. The wearable was Fitbit tracker.

Lancet Digit Health

  • Search in PubMed
  • Search in NLM Catalog
  • Add to Search doi: 10.1016/S2589-7500(19)30222-5. Epub 2020 Jan 16.
  • . 2020 Feb;2(2):e85-e93.

Harnessing wearable device data to improve state-level real-time surveillance of influenza-like illness in the USA: a population-based study

Jennifer M Radin  1 , Nathan E Wineinger  2 , Eric J Topol  2 , Steven R Steinhubl  2

This manuscript is a resubmission of an earlier submission. The following is a list of the peer review reports and author responses from that submission.

Round 1

Reviewer 1 Report

Unfortunately, I don't believe the content of the manuscript is enough for a scientific study.

Reviewer 2 Report

This case report is interesting. However, there were many issues that should be addressed before further consideration.
1. Please update the epidemiological data of the COVID-19 pandemic.
2. Please use the same medical terms throughout the manuscript. Please consider SARS-coV-2 instead of 2019-nCoV.
3. The abstract should focus on the unique characteristics of this case and the importance of consumer sleep wearable monitoring of sleep problems.
4. Please watch for the abbreviation using as author guidelines.
5. The case description of the Introduction should be moved to the Materials and Methods.
6. The definition of the Covid period is not clear. Do you mean SARS-coV-2 infection?
7. Please introduce detailed information on the three wearable devices. Which one can confirm the sleep stage (such as EEG signals)? What are the senor types? What are the detection algorithms? How about the precisions of these wearables?
8. The logistic regression models are not necessary. This is a single case study. All the statistical analyses are not reasonable. May you consider presenting the changes of the TST, WASO, HR, and skin temperature. 

Reviewer 3 Report

This manuscript reports the first case of the continuous sleep study ( by validated wearables) in COVID-19 patient quarantined at home. This study detected that sleep is specifically affected during COVID.
This manuscript shows a case report.
The following comment should be considered to enhance the quality of the manuscript:
1. The quality of the figures is poor. Especially, Figs. 1B(a) and 2(c)
2. It could be interesting to present additional details of the consumer sleep wearables (CSW), at least including a diagram or a flowchart. These additional details will illustrate the measurement system; I think that the illustration of these details will increase the interest of the reader board of the sensors journal.